# A Simplified Facility Management Tool for Condition Assessment through Economic Evaluation and Data Centralization: Branch to Core

Giovanna Acampa * and Alessio Pino *

Faculty of Engineering and Architecture, University of Enna "Kore", 94100 Enna, Italy
* Correspondence: giovanna.acampa@unikore.it (G.A.); alessio.pino@unikorestudent.it (A.P.);
  Tel.: +39-3482-548596 (A.P.)

**Abstract:** The field of facility management, especially concerning condition assessment, is affected by two main issues: one is the incompleteness and heterogeneity of information transfer between the involved subjects; the other is the frequent lack of specific advanced skills needed for technically complex tools. The immediate consequences of this process inefficiency fall on economic and environmental aspects: the unavailability or incorrect structuring of data related to building conditions does not allow for making optimal choices concerning interventions on components. This paper attempts to provide a solution in this framework by presenting a methodology for simplified condition assessment, in which the evaluation of decay parameters draws from economic evaluation techniques, and which optimizes data collection, systematization, and elaboration, also integrating it with a mobile app for automatic data upload and centralization. The research underlying its development draws from decay evaluation criteria and national standards for the analysis and breakdown of buildings. The methodology was tested on a case study of the Cloister of Santa Croce in Florence, which also served as the client of the tool. The proposed methodology stands as an easily implementable integration to condition assessment for maintenance planning and building inspection activities.

**Keywords:** facility management; maintenance planning; building inspection; app; economic evaluation; decay evaluation; ISO 12006-2:2015

## 1. Introduction

In building construction, the management of an artifact represents the longest-lasting and most relevant phase in its whole life cycle [1–3]. This relates to the impacts associated with the resources consumed for its maintenance, which can significantly vary according to the choices that are performed concerning interventions, restorations, and operations. These variables can be intended in at least three dimensions: economic impacts [4], which are caused both by the materials employed for substitutions and repairs [5], and by the equipment, tools, and personnel required for the execution of interventions and inspections [6]; environmental impacts, associated not only with the consumption of materials and resources that is embodied in the environmental footprint [7,8], but also with the transportation processes involved by all maintenance operations [9]; and the effects on the usability of the building caused by its downtime [10], which is of course affected by the temporal distribution of interventions and their timing [11]. This aspect is particularly relevant for businesses and public buildings [12], and of course benefits from the synergic execution of interventions on different components; however, the combination of different operations should be pondered in relation to their different states of conservation [13], without anticipating or delaying interventions excessively with respect to the timing suggested by their durability aspects [14].

Dealing with the best choices in relation to these criteria and areas of impact means optimizing maintenance interventions [15], hence representing a complex task, due to

the variety of trade-offs and parameters involved, in addition to the probabilistic and multiparametric nature of building component performance. In other words, this factor does not allow for making maintenance evaluations only based on the knowledge of physics and dynamics of performance, instead forcing a reliance on the collection of information on each component over time; that is, the possibility to establish correct and optimized planning requires correct and updated knowledge regarding the evolution of the conservation state of building components.

This refers to the specific aspect of building condition assessment, which is the analysis of the condition of a building, regarding design characteristics, age, employed materials, construction typologies, and the conservation state [16].

The aspects of knowledge and knowledge updating have a peculiar and critical role in building construction [17], mainly in relation to two elements:

- The codification of functional databases for the storage of all of the data needed to identify and define building components, including all information required for their maintenance activities and conservation state [18];
- The combination and overlapping of different figures involved in the collection, visualization, consultation, and employment of building-related information, characterized by different levels of skills and possibilities for interaction with complex classification criteria, as well as different typologies of data input.

These two aspects converge toward the major issue of data fragmentation and information loss: the former is intended as the need for a data collection output system that can host all of the required data and filter the useful information in order to achieve a compact interface, while the latter represents the main cause for the presence of different documents—inspection reports—with diversified parameters and lexicons [19], which lead to an inconsistent information framework.

This is the picture in which this paper fits. The work that is presented here has been developed within the WARMEST project, an R&I Staff Exchange (RISE) project funded by the European Union in the framework of the Horizon 2020 program, concerning the facility management of buildings and availing of economic assessment techniques from the field of appraisal [20] to fine-tune:

- A matrix template for the collection of all information on building components and their maintenance aspects, supported by the technique of work breakdown structure [21];
- An easily implementable open-access and open-source app to enable fast data collection and upload from maintenance inspections directly to the general database, both limiting the amount of unrequired information and optimizing the intuitiveness and orientation of data input;
- An automatic process for the definition of indexes related to the evaluation of the state of decay of building components, based on the data collected from inspection processes.

This methodology has been developed within the WARMEST project as "FM Branch to Core"; this name is related to its capacity and base concept to automatically link peripherally collected information with the central database. The description of its components and the theoretical aspects behind its structure and layout will be followed by its synergic application.

The research for the development of this methodology was requested by the Technical Department of the Basilica di Santa Croce in Florence, Italy. Their needs consisted of two items:

- the standardization of decay descriptors for monitoring activities, and a framework for their definition;
- the simplification of monitoring procedures and the collection of the related data.

The indication of these goals oriented the development of the methodology, and the cloister of Basilica di Santa Croce represented the case study for the implementation of the tool. Indeed, its architectural value, the high number of components, and the

significant effects produced by the optimization of its maintenance actions represent a feasible litmus test for the validity of these tools, and the benefits of their exportation to common maintenance practice.

## 2. Materials and Methods

### 2.1. Literature Review

Given the wide range of benefits that correct maintenance planning provides [22], various methodological frameworks for building condition assessment (BCA) have been developed throughout the last 20 years [23], including opportunities for the digitization of the BCA process: one example is Docu-tools[®] [24], which is based on the ex post digitization of building documents, including inspection reports. However, it does not deal with the issue of lack of standardization among the various assessment scales. Serrat et al., in an interesting research project from 2020 [25], develop a platform titled, "*Full Interactive Visualization Method for Building Condition Assessment*", in which information on the state of decay of building components is directly obtained from their survey, through UAV photogrammetry. This methodology allows data systematization and automated processing, but aspects related to decay quantification and parametrization are only related to geometric and colorimetric factors. Matos et al. [26] propose the integration of BCA with BIM to enable improved representation features and visualization of the whole building that the single assessed components are part of.

A significant contribution to the range of platforms for building condition assessment derives from facility management software companies: some of the most diffused ones are AkitaBox[®] [27], FacilityForce[®] [28], and Hippo CMMS[®] [29]. Hippo CMMS also provides a mobile app for keeping track of inspections and interventions and scheduling preventive maintenance. However, they all share the same issues: they can only be used through a paid subscription, which is hardly suitable for small construction companies and administrations: the most critical category for the upgrade of methods and processes [30]. Moreover, their algorithms are not customizable, so the evaluation of decay has to either be performed through software parameterization rules or manual input.

### 2.2. Work Breakdown Structure

While the expression work breakdown structure appears in the ISO 21511:2018 standard and refers to a project management tool that takes a step-by-step approach to completing large projects with several moving pieces, its specification for the hierarchization of buildings and assets is defined by the ISO 12006-2:2015 standard, in "Building construction—Organization of information about construction works Part 2: Framework for classification". That section is aimed at defining a reference framework for the development of classification systems for the built environment, listing several recommended titles and tables, to be used for specific classes of information. However, it does not provide operative guidelines to realize a model for the subdivision of building components; instead, this type of systematization [31] is present in national, local, and European codes, ranging from the American OmniClass, MasterFormat, and UniFormat, to the EU CEEC building classification, the UK UniClass, the Swedish CoClass, the Danish CCS, and the Italian system, coded by UNI 8290.

### 2.2.1. Classification According to UNI 8290 Standard

The Italian code, which will be analyzed here in depth, was published in 1981 and still stands as a reference on the Italian territory. It consists of three parts and one update:

- UNI 8290-1: Residential building. Building elements. Classification and terminology;
- UNI 8290-2: Residential building. Building elements. Analysis of requirements;
- UNI 8290-3: Residential building. Building elements. Analysis of agents;
- UNI 8290/1 FA 122-83: Residential building. Building elements. Classification and terminology.

The code suggests a subdivision of the building system into multiple levels, with homogeneous rules, which draw from UNI 7687 Section 4. In particular, the base classification has three different levels:

- Technological unit class ("Classe di unità tecnologica"), which is a grouping of homogeneous building elements by physical and functional continuity;
- Technological unit ("Unità tecnologica"), the set of technical elements that perform functions to fulfill users' needs;
- Technical element class ("Classe degli elementi tecnici"), the class of products that performs different specific functions within one, or more, technological class.

The following, Table 1, reports the structure of the WBS (first three levels), carried out according to the UNI 8290 standard.

**Table 1.** Essential levels of the work breakdown structure, according to UNI 8290 standard.

| Technological Unit Classes | Technological Units | Technical Element Classes |
|---|---|---|
| 1. Load-bearing structure | 1.1. Foundation structure | 1.1.1. Shallow foundation structure |
| | | 1.1.2. Deep foundation structure |
| | 1.2. Aboveground structure | 1.2.1. Horizontal aboveground structural elements |
| | | 1.2.2. Spatial aboveground structural elements |
| | 1.3. Retaining structure | 1.3.1. Vertical retaining structural elements |
| | | 1.3.2. Horizontal retaining structural elements |
| 2. Enclosure | 2.1. Vertical enclosure | 2.1.1. Vertical perimetral walls |
| | | 2.1.2. Vertical windows and external doors |
| | 2.2. Bottom horizontal enclosure | 2.2.1. Ground floor slab |
| | | 2.2.2. Horizontal windows |
| | 2.3. Horizontal enclosures versus external spaces | 2.3.1. Floor slabs on open spaces |
| | 2.4. Top closure | 2.4.1. Roof slab |
| | | 2.4.2. Horizontal windows and external doors |
| 3. Internal partition | 3.1. Vertical internal partition | 3.1.1. Vertical partition walls |
| | | 3.1.2. Internal doors and windows |
| | | 3.1.3. Protection elements |
| | 3.2. Horizontal internal partition | 3.2.1. Floor slabs |
| | | 3.2.2. Lofts |
| | | 3.2.3. Internal horizontal windows |
| | 3.3. Sloped internal partition | 3.3.1. Internal stairs |
| | | 3.3.2. Internal ramps |
| 4. External partition | 4.1. Vertical external partition | 4.1.1. Protection elements |
| | | 4.1.2. Separation elements |
| | 4.2. Horizontal external partition | 4.2.1. Balconies and lodges |
| | | 4.2.2. Boardwalks |
| | 4.3. Sloped external partition | 4.3.1. External stairs |
| | | 4.3.2. External boardwalks |

**Table 1.** *Cont.*

| Technological Unit Classes | Technological Units | Technical Element Classes |
|---|---|---|
| 5. Technological supply systems and services | 5.1. Air conditioning system | 5.1.1. Supply |
| | | 5.1.2. Thermal units |
| | | 5.1.3. Fluid treatment units |
| | | 5.1.4. Distribution networks and terminals |
| | | 5.1.5. Condensate drain networks |
| | | 5.1.6. Exhalation pipes |
| | 5.2. Water supply system | 5.2.1. Connections |
| | | 5.2.2. Hydraulic machinery |
| | | 5.2.3. Storages |
| | | 5.2.4. Heating devices |
| | | 5.2.5. Cold water distribution networks and terminals |
| | | 5.2.6. Hot water distribution networks and terminals |
| | | 5.2.7. Hot water re-circulation networks |
| | | 5.2.8. Sanitary appliances |
| | 5.3. Sewage disposal system | 5.3.1. Wastewater sewage networks |
| | | 5.3.2. Greywater sewage networks |
| | | 5.3.3. Rainwater sewage networks |
| | | 5.3.4. Secondary ventilation networks |
| | 5.4. Gas disposal system | 5.4.1. Supply |
| | | 5.4.2. Machinery |
| | | 5.4.3. Channeling networks |
| | 5.5. Solid disposal system | 5.5.1. Dropping pipes |
| | | 5.5.2. Exhalation pipes |
| | 5.6. Gas supply system | 5.6.1. Connections |
| | | 5.6.2. Distribution networks and terminals |
| | 5.7. Electric system | 5.7.1. Supply |
| | | 5.7.2. Connections |
| | | 5.7.3. Electric machinery |
| | | 5.7.4. Distribution networks and terminals |
| | 5.8. TLC system | 5.8.1. Supply |
| | | 5.8.2. Machinery |
| | | 5.8.3. Distribution networks and terminals |
| | 5.9. Fixed elevation system | 5.9.1. Supply |
| | | 5.9.2. Machinery |
| | | 5.9.3. Movable parts |

**Table 1.** *Cont.*

| Technological Unit Classes | Technological Units | Technical Element Classes |
|---|---|---|
| 6. Safety system | 6.1. Fire protection system | 6.1.1. Connections |
| | | 6.1.2. Sensors and transducers |
| | | 6.1.3. Distribution networks and terminals |
| | | 6.1.4. Alarms |
| | 6.2. Grounding system | 6.2.1. Collection networks |
| | | 6.2.2. Earth plates |
| | 6.3. Lightning protection system | 6.3.1. Rods |
| | | 6.3.2. Network |
| | | 6.3.3. Earth plates |
| | 6.4. Anti-theft and anti-intrusion system | 6.4.1. Supply |
| | | 6.4.2. Sensors and transducers |
| | | 6.4.3. Network |
| | | 6.4.4. Earth plates |
| 7. Internal equipment | 7.1. Home furniture | 7.1.1. Container walls |
| | 7.2. Service block | 7.1.2. Service block |
| 8. External equipment | 8.1. Collective external furniture | 8.1.1. Collective external furniture |
| | 8.2. External fixtures | 8.2.1. Fences |
| | | 8.2.2. External pavement |

The decomposition of building systems can be compounded by the addition of further levels, customizing it according to the needs of the analysis and the characteristics of the considered building. In other words, if the single technical elements do not have significant differences, or if such differences are not the object of diversified actions, the subdivision of the building can be stopped at the determination of the classes of technical elements. Otherwise, it is possible to add specifications for the single technical elements, attributing to them an ID identifier, and classifying them according to their material, performance variables, location in the building, and any other kind of data, including information related to maintenance activity.

*2.3. Methods for the Evaluation of Building Condition*

Building condition is generally evaluated through qualitative and quantitative criteria, which are often incorporated into multi-parametric assessments [32]. Recently, quality evaluation systems and building certification protocols are being utilized in an increasingly wide and rich application scope, leading to their numerical increase and diversification; among them, BREEAM and LEED [33,34] are some of the best-known ones. These are based on sets of indicators with associated scores and weights for each aspect of building quality, which are then combined to determine the rating of the analyzed building. Of course, they also include indicators for the technical characteristics of the components, providing scales of scores to be attributed according to the evaluators' judgment; as highlighted by several researchers [35–39], these represent a viable basis for the evaluation of component performance at a given time over the life cycle of buildings, concerning qualitative evaluations. Indeed, they are based on condition scales, which involve a linguistic representation, articulated around various qualifiers (protected/exposed; good/poor; slight/severe decay; and satisfactory/bad). The purpose of building certification protocols is inherently different from that of building condition assessment methods, as they are intended to assess the general state of the building for reasons related to energy performance; however, the

parameters and scales adopted to evaluate this specific aspect within them can be taken as a reference among qualitative approaches for the determination of building condition.

Quantitative indicators, as a result, are more useful for providing an objective computational framework for evaluating the performance of building components, and the scientific literature presents several of them, distinguished according to the typology of parameters that they evaluate (i.e., economic, technical, functional, and user satisfaction). Among technical aspects, the following are the most relevant:

- Building performance indicator (BPI) [40], based on the quantitative expression of the physical and functional condition of the building;
- Building efficiency index (BEI) [41], related to energy performance, which includes building services, occupant comfort, and climate conditions in the evaluation;
- Degradation index (A) [42], related to the degradation of technological systems;
- Level of service (LOS), which measures technological performance in relation to environmental quality;
- Environmental condition (EC), based on environmental performance.

Most of these formulations are oriented toward the evaluation of a whole set of components, as most of their assessed parameters involve a multiplicity of functions and associated technical elements. As an example, energy performance cannot be evaluated by considering a single component, but must instead refer to all components participating in the fulfillment of a given energy need. Instead, the degradation index appears to be particularly useful for the obtainment of a parameter referring to a single technical element, as it can allow the performance of specific evaluations on the convenience and optimization of the maintenance activity on a "minimal unit" of the building, by the WBS logic that was described above.

Degradation Index

The degradation index [42] is proposed here in a customized synthetic expression, represented by the following equation:

$$i_d = \frac{A_d}{A_t} \cdot K_d \cdot K_b \tag{1}$$

where:

- $i_d$ is the degradation index;
- $A_d$ is the area of the technical element affected by decay;
- $A_t$ is the total area of the surface of the technical element;
- $K_d$ (decay severity) is a factor considering the severity of the decay;
- $K_b$ (decay burden) is a weighting factor reflecting the impact of decay, in relation to the cost of its associated restoring intervention.

The definition of the terms $K_d$ and $K_c$ is part of the research presented here, as their attribution of tabulated values is an original contribution. The modeling of the former requires considering and quantifying the impact of each degradation phenomenon on the state of conservation of the building component that it affects. Concerning the latter, it is necessary to collect data on the most suitable restoring interventions associated with each typology of decay and establish the values of the coefficient in relation to them.

### 2.4. Classification of Decay Typologies

Decay typologies for building components are described in several national and international norms. The Italian national framework—with reference to UNI norms—provides classifications for several different materials; in particular, the UNI 11182:2006 norm codifies decay typologies for stone elements, while decay typologies for wood materials are listed in the UNI 11130:2004 code. Concerning metallic elements, the international ISO 12944 code can be used as a reference; however, it does not define different typologies of

decay, but instead retraces all forms of performance decay for metallic elements to the loss of material, quantifying its entity and evolution by identifying various classes.

### 2.4.1. Stone Elements

UNI 11182:2006 identifies two classes of decay typologies, distinguishing them into alterations and degradations. The main difference between the two categories is that the forms of decay in the former, unlike the latter, do not imply a worsening in the performance of the component. This refers to modifications such as patina or chromatic alteration; however, this subdivision only considers technical performance. Instead, it has to be taken into account that the aesthetic aspect plays a key role in sites with architectural and artistic relevance and defects affecting it must be analogously cured. Hence, these two categories are grouped and described together. With reference to stone elements, the decay typologies listed in the UNI 11182:2006 code are reported in Table 2.

**Table 2.** List of decay typologies for stone elements, according to UNI 11182:2006 code, and their synthetic descriptions.

| Decay Typology | Description |
| --- | --- |
| Chromatic alteration | Natural, diffuse variations of color parameters. |
| Alveolization | Presence of cavities with variable shapes and sizes, called alveoli. |
| Leaking | Vertical trace, or often a set of parallel traces. |
| Biological colonization | Visible presence of micro- and/or macro-organisms (algae, fungi, moss). |
| Crust | Superficial modification of the stone element, with variable thickness and higher hardness than the rest of the surface. |
| Deformation | Variation of the line or form that interest the whole material. |
| Differential degradation | Loss of material from the surface, highlighting the heterogeneity of the surface and structure. |
| Superficial deposit | Accumulation of extraneous materials of different nature, among which powder, soil, etc. |
| Powdering | Decohesion and fall of the material, in the form of powder or fine fragments. |
| Detachment | Loss of continuity between plaster layers, both among each other and with the substrate. |
| Efflorescence | Crystal- or powder-shaped, generally white superficial formation. |
| Erosion | Loss of material from the surface, which generally appears as compact. |
| Exfoliation | Formation of one or more thin laminar areas named folia. |
| Crack | Loss of continuity in the material, leading to the reciprocal movement of the single parts. |
| Capillary rise front | Boundary of water capillary rise, manifested by the formation of efflorescence and/or material loss. |
| Graffiti | Undesired apposition of colored paint on the surface. |
| Lacuna | Loss of continuity of surfaces. |
| Stain | Localized chromatic variation on the surface, correlated both to natural components of the material and to extraneous elements. |
| Lack | Loss of tridimensional elements (an arm of a statue, a handle of an amphora, a part of a bas-relief). |
| Patina | Natural modification of the surface, which cannot be related to decay phenomena. |
| Biological patina | Thin and homogeneous layer, mainly constituted by micro-organisms, which can vary by consistency, color, and adhesion to the substrate. |

**Table 2.** *Cont.*

| Decay Typology | Description |
|---|---|
| Film | Transparent or semi-transparent superficial layer of substances, which are coherent to each other and do not belong to the stone element. |
| Pitting | Formation of numerous and adjacent blind holes. |
| Presence of vegetation | Presence of grass, shrubs, or trees. |
| Bulging | Localized superficial lifting of material, with variable form and consistency. |
| Scaling | Presence of irregular parts with consistent and variable thickness, called scales. |

### 2.4.2. Wood Elements

The UNI 11130:2004 code establishes two main typologies of decay for wood elements—biotic and abiotic—depending on whether degradation is determined by animal and vegetal species, or by mechanical, chemical, and physical causes. For wood, the main causes of degradation are related to water, which produces humidity in the whole section, and the action of bugs that feed themselves with wood. These two factors lead to a heterogenous framework of decay susceptibility, depending on the location of the building and its exposition to the sea or specific entomologic fauna. Table 3 reports the decay typologies indicated in the UNI 11130:2004 code.

**Table 3.** List of decay typologies for wood elements, according to UNI 11130:2004 code, and their synthetic descriptions.

| Decay Typology | Description |
|---|---|
| Bug attack (biotic) | Degradation caused by xylophagous bugs, which dig galleries inside the wood, leaving characteristic holes on the surface. |
| Cavity (biotic) | Degradation caused by fungi, which produce a progressive loss of mass, resistance, hardness, and variations of color and aspect. |
| Attack from marine organisms (biotic) | Occurs in seawater due to the action of xylophagous crustaceous and mollusks, which dig galleries inside wood or on its surface. |
| Bacteria attack (biotic) | Pieces of wood affected by high humidity for a long time (or immersed in water) show a darker and softer surface. |
| Photolytic attack (abiotic) | Surface degradation due to photo-oxidative processes. |
| Chemical attack (abiotic) | Due to oxidizing acids, bases, and agents. |
| Hydrolysis (abiotic) | Degradation due to the contact with acid or basic aqueous solutions, leading to the asportation of soluble substances from wood. |
| Petrification or mineralization (abiotic) | Substitution of organic substances in wood with inorganic substances, due to the deposit of mineral substances. It can be caused by contact with oxidizing metals. |
| Roasting (abiotic) | Preliminary phase of thermal degradation, in which wood suffers chromatic alteration and reduction of hygroscopicity and mechanical characteristics. |
| Carbonation (abiotic) | Degradation process caused by the lack of oxygen, which leads to the transformation of wood into carbon. |
| Combustion (abiotic) | It can be slow (pyrolysis) or fast (in case of fire). |

### 2.4.3. Metallic Elements

Unlike the other codes, the ISO 12994 code does not provide a classification of decay typologies. In this code, the predominant decay typology for these components is considered to be corrosion, which is assimilated to a loss of material from the element. Concerning it, the following factors are classified:

- The level of corrosion after which the first intervention should be performed, in accordance with indications from the ISO 4628 code, and divided into 4 classes;
- The corrosiveness of the environment in which the metallic element is located, which affects the evolution of decay over time, and is divided into 6 classes for aboveground structures, and 4 classes for underwater or underground structures.

### 3. Description of the Methodology

The methodology, FM Branch to Core, presents the following additions to the traditional paradigm of building monitoring and management:

- Adoption of a specific WBS system that includes all data required for the individuation of the characteristics of each component in relation to maintenance activity, called Master Database;
- Application of an algorithm to determine the intervention to be performed on each component following the calculation of its degradation index, and the consequent choice of the most suitable maintenance intervention to associate with them;
- Introduction of an optimized system for the automatic update of the characteristics regarding the conservation state of each component through a dedicated app, to be used by all the technicians involved in inspection activities.

These actions fit within the traditional framework of maintenance activities, whose technique is currently evolving and is based on the planning and execution of inspections and interventions. Inspections are meant to provide updated data on the state of decay of components, hence adjusting maintenance plans to comply with the need to intervene on them.

Figure 1 graphically outlines the research carried out for the development of the Branch to Core method and app, presented for the first time in this paper. As shown here, the development of the methodology did not cover the analysis of the relationships between decay conditions and recommended maintenance interventions, which stands as the operational part of facility management; instead, the focus has been on building a systematized and reliable data collection system as a base for the determination of suitable maintenance interventions.

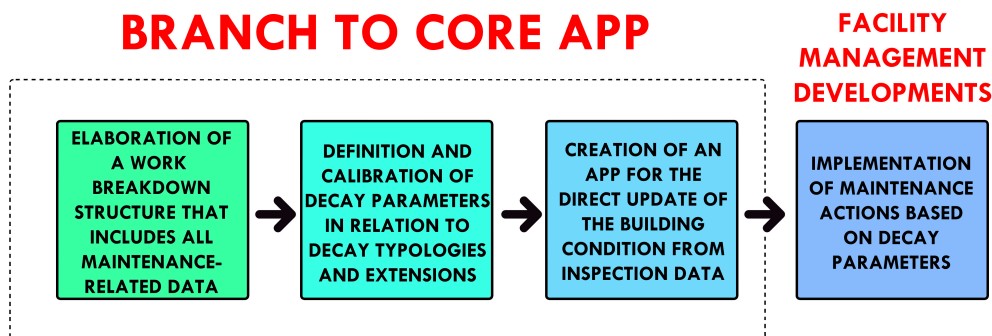

**Figure 1.** Workflow of the Branch to Core App, whose methodological framework includes the steps within the dotted line; the fourth step is the further development that it provides the basis for.

### 3.1. Master Database

The WBS layout recommended by the UNI 8290 code, outlined in Section 2.1, is based on the progressive decomposition of a unitary system—the building—into an increasing number of instances, corresponding to the further, more detailed, level of analysis (technological unit classes, technological units, technical element classes, or technical element). In

accordance with IT databases, this approach was reversed by focusing the attention on the single technical elements, rather than on the system, hence turning the upper levels of the WBS into attributes of the technical elements. In addition to that, further levels of attributes were defined for each technical element. These include intrinsic characteristics required for the analysis of maintenance-related aspects of each component—material, location, and exposure—characteristics related to the state of conservation, which change over time and need to be updated—decay typology and decay severity—and characteristics related to the maintenance activity to be performed—maintenance and inspection costs, frequencies, and dates.

The result is the Master Database, a database whose columns, attributed to each of the components, are the following items, subdivided into levels:

1. Level 1:

   Technological Unit Class;

2. Level 2:

   Technological Unit;

3. Level 3:

   Technical Element Class;

4. Level 4:

   Technical Element;

5. Level 5:

   Material;
   Exposure;
   Position;
   Surface Area;

6. Level 6—Decay Evaluation:

   Decay Typology;
   Decay Severity;
   Decay Burden;
   Decay Area;
   Degradation Index;

7. Level 7—Maintenance Plan:

   Inspection typology;
   Inspection description;
   Inspection frequency;
   Maintenance intervention typology;
   Maintenance intervention description;
   Maintenance intervention frequency;
   Last inspection date;
   Last maintenance intervention frequency;
   Inspection costs (€/h);
   Maintenance costs (€/m$^2$).

The database is implemented as a Microsoft Excel worksheet, hence each technical element corresponds to a row and each entry within the levels is a column.

In Level 4, the Technical Element ID is a code with 4 entries, corresponding to: the number of the technological unit class; the number of the technological unit; the number of the technical element class; the number attributed to the specific technical element (a column, a window, or the plaster of a wall, for example).

Data from Level 1 to Level 5 are attributed to each of the components (technical elements) when the database is created for the specific building that is the object of the

maintenance planning. Following this, data for Level 6 are inserted with the first survey and updated at every subsequential one. Finally, data from Level 7 are semi-automatically obtained from the data in Level 6 according to the value of the degradation index for each specific component: indeed, the degradation index is calculated based on the decay typology and decay severity, as illustrated in Section 2.2.1. The list of inspections and interventions, together with the associated data, which are recalled for the other entries, is contained in an external attached database, the "Maintenance Archive".

Maintenance Archive

The Maintenance Archive is an additional worksheet in the same Microsoft Excel file as the Master Database. It contains a set of maintenance cards for inspections and maintenance interventions, providing the data required by Level 7 of the Master Database, in addition to the raw data through which those are calculated, and the applicability conditions in relation to the thresholds of the maintenance index. The data for each card in the Maintenance Archive are the following:

- Code of the Maintenance Card;
- Technical element class;
- Typology of inspection/intervention;
- Description of the inspection/intervention;
- Reference Price List;
- Reference Item Code(s) from the original Price List;
- Reference Item Description(s) from the original Price List;
- Reference measure unit of the Item(s) from the original Price List;
- Unit cost(s);
- Total cost.

The maintenance cards are built up by compounding items from price lists, including single operations or materials that are needed to perform the overall action. Of course, the reference price list, indicated on each maintenance card, stands as a criterion for the applicability of the inspection/intervention cards in the specific maintenance plan, as it is necessary to consider the costs in the local economic and productive context. For this reason, the Maintenance Archive must be integrated and adapted according to the geographic location, and national values can be used only in the lack of regional ones (with reference to the Italian case).

Figure 2 shows an example of a maintenance intervention card in the Maintenance Archive.

| 01.04.01.I02 (A) | | | Wood structures (B) | | |
|---|---|---|---|---|---|
| Restoration of bolt joints and metal connections (C) | | | Restoration of the anti-rust protection of metal elements by removing rust and applying protective paint. Repair of corrosions and cracks with local wielding through connection elements (D) | | |
| Prezzario DEI Recupero Restauro Manutenzione (2020) (E) | | | 10.91 € (J) | | |
| B65064.a (F) | Preparation of windows and ironworks, by sanding and cleaning with a wire brush. (G) | | | sqm (H) | 2.31 € (I) |
| B65065.a (F) | Opaque-surface rust protection primer, applied with a brush, up to 3 cm in diameter or side. (G) | | | sqm (H) | 1.40 € (I) |
| B65066.a (F) | Lead-based rust protection primer, applied with a brush on prepared surfaces, for windows and ironworks. (G) | | | sqm (H) | 7.20 € (I) |

**Figure 2.** Maintenance Card "01.04.01.I02". The entries in the card are indicated with the letters in brackets: code of the maintenance card (A); technical element class (B); typology of intervention (C); description of the intervention (D); reference price list (E); reference item code (F); reference item description (G); reference measure unit of the item (H); reference quantity; unit cost (I); total cost (J).

Each maintenance intervention card is associated with a value of the degradation index of the related technical element class, in correspondence with which the intervention that should be performed. Hence, the algorithms that will be shown in the following,

Section 3.2, lead to the determination of the most suitable interventions according to the value of the Degradation Index.

### 3.2. Degradation Index Algorithm

The formula from the scientific literature reported in Section 2.2.1 is not paired with a standardized process for the evaluation of the involved parameters. In this work, specific formulas are defined for each of the terms of the equation, based on national and international codes, consultation from experts in the fields of maintenance and management, and economic evaluation. With reference to the formula from the scientific literature reported in Section 2.2.1, the original addition which allows the implementation of the algorithm is represented by:

- ranking of decay typologies (exemplified for stone elements) according to the level of degradation that they cause to the component that they affect for the determination of $K_D$;
- evaluation of the economic impact of decay typologies in relation to the interventions that they imply, based on the consultation and analysis of several national and regional price lists, for the determination of $K_B$.

#### 3.2.1. Decay Severity

Decay severity ($K_D$) is automatically calculated, depending on the decay typology, on a scale from 0 to 4: 0 corresponds to "No Decay", hence to the absence of the indication of any decay typology, while scores from 1 to 4 range from "Slight Decay" to "Medium Decay", "Strong Decay", and "Severe Decay". Moreover, scores from 3 to 4 imply the loss of material from the component.

Table 4 shows the repartition of all Decay Typologies into these four severity classes.

**Table 4.** Repartition of decay typologies of stone elements into the four severity classes.

| Slight Decay (1) | Medium Decay (2) | Strong Decay (3) | Severe Decay (4) |
|---|---|---|---|
| Chromatic alteration | Biological patina | Alveolization | Deformation |
| Leaking | Crust | Differential degradation | Detachment |
| Graffiti | Superficial deposit | Powdering | Exfoliation |
| Stain | Efflorescence | Erosion | Bulging |
| Patina | | Crack | Scaling |
| Film | | Presence of vegetation | |
| | | Pitting | |
| | | Capillary rise front | |
| | | Lacuna | |
| | | Lack | |

#### 3.2.2. Decay Burden

$K_B$ coefficients correspond to the ratio between the cost of the interventions required to restore a component that is affected by a given typology of decay, and the cost of the substitution of the component. The evaluation of this parameter is strongly rooted in economic evaluation techniques: it is based on the conceptual assumption that maintenance priority should be defined also in relation to the repair costs entailed by each decay typology. These are not directly proportional to the severity classes defined in Section 3.2.1, as they depend on the typology of repairing intervention and the related cost.

The values related to the intervention costs have been drawn and averaged from the following price lists:

- Price List of Public Works of Tuscany Region 2020 (Section 3—restoration);
- Price List of Public Works of Piedmont Region 2020 (Section 27—conservation and restoration of cultural heritage);
- Price List of Public Works of Marche Region 2019 (Section 5—reinforcement and restoration works);

- Price List of the Superintendence for Archaeology, Fine Arts, and Landscape of the Municipality of Venice and its Lagoon—General Restoration Works 2018;
- Price List for the Conservation and Restoration of the Cultural and Landscape heritage of Calabria 2019;
- DEI Price List for the restoration of Artistic Heritage (2019);
- DEI Price List for the restoration, restructuring, and maintenance (2020).

This operation of comparison and analysis between the costs of repair and substitution was performed by considering four typologies of intervention for each component, associated with the classes of decay typologies listed in the previous Section. Indeed, decay typologies belonging to the same severity class are treated with the same typology of restoring intervention in maintenance practice, and so it is a valid approach to group and unify them for this purpose. Table 5 reports the $K_C$ for each class of decay typology, based on the unit cost of the related restoring intervention.

**Table 5.** Attribution of $K_B$ coefficients to the decay typologies, grouped by severity class, for stone elements, with indications of the associated interventions.

| Severity Class | Restoring Intervention | Intervention Unit Cost (€/sqm) | $K_C$ |
|---|---|---|---|
| 1 | Simple Cleaning | 50.54 | 0.18 |
| 2 | Complex Cleaning | 61.78 | 0.22 |
| 3 | Material Integration | 213.41 | 0.76 |
| 4 | Material Reinforcement | 481.89 | 1.71 |

Through this modeling, the Master Database automatically reports the calculation of the degradation index, based on the indication of one, or more, decay typology, and its/their extension. Since the semi-automatic procedure involves the maintenance planner's active decision, in case of multiple decay typologies on the same component leading to a high economic burden, the convenience of performing several restoring interventions versus substituting the component will be evaluated before the definition of the maintenance plan.

### 3.3. Branch to Core App

The Branch to Core App is a mobile app for data input into a central database from mobile and pc devices. The data input can be performed only on specific, pre-selected fields, and all of the entries have drop-down lists to prevent technicians from inserting data that do not comply with the classifications and groupings that have been established in the other steps of the methodology. Moreover, the front-end interface is also limited to the data that can be useful to technicians during inspection operations, in order to identify the components according to their classification within the WBS of the Master Database, hence taking all precautions to avoid information clashes and inconsistency throughout the whole process.

The Branch to Core App was created through AppSheet (Available online: https://www.appsheet.com/Home/Start, last accessed on 25 February 2023), a no-code development platform that enables the creation of mobile apps, which can then be launched and used from mobiles and tablets, using a data source from a shared online database. In this methodology, Google Drive (Google, Mountain View, CA, United States) has been chosen as cloud storage for the Master Database; the main reason for that is Google acquired AppSheet in 2020, hence its related cloud service presents the most functional compatibility features with the platform, and this combination will presumably receive optimization updates in the future.

AppSheet automatically creates an app interface from a spreadsheet, which can be edited to specify which fields can be viewed or modified from mobile devices, and whether and how it is possible to add new entries to the database. The data source has a dynamic relationship with the app, as all the data that are introduced or edited from mobile devices are automatically also changed in the spreadsheet in the cloud.

It is worth noting that for each building that is the object of a maintenance plan, it is necessary to create a separate Branch to Core App, just as each Master Database is unique to a building. Figure 3 shows some screenshots of its use.

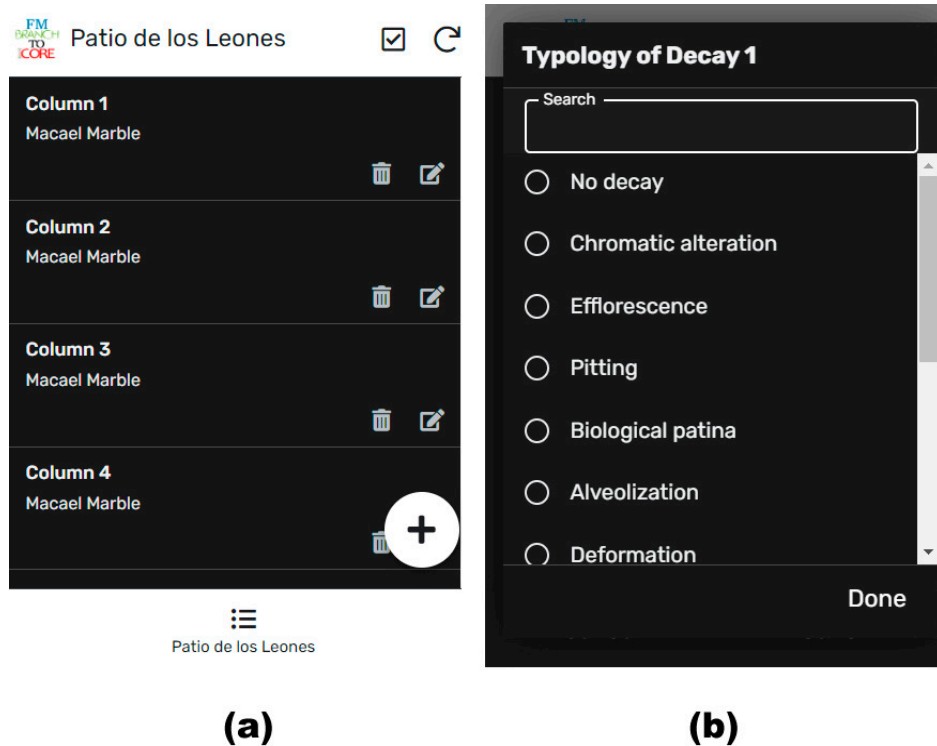

(a)          (b)

**Figure 3.** Screenshots from the Branch to Core App for Patio de Los Leones in the monumental complex of Alhambra, Granada, Spain: (**a**) menu overview of column components, with the indication of the technical element and the material; (**b**) drop-down menu for the input of the typology of decay for a single component.

Based on the WBS setting of the Master Database, before the definition of the architecture of the Branch to Core App, it was necessary to define which fields should appear in the app view, which ones can be edited, and, in relation to that, the set of information from which each entry could be chosen. Table 6 reports the outline of the data that can be viewed from the Branch to Core App and the associated conditions for the input/modification of the editable ones.

As shown in Table 6, maintenance operators only have to fill in $4 + 2n$ entries for each building component, where $n$ is the number of observed decay typologies. For better comprehension, the four fixed entries are as follows: inspection typology, intervention typology, last inspection date, and last intervention date; decay typology and decay area have to be filled in for each form of decay. Hence, this makes the process particularly smooth and reliable; as repeatedly remarked, one of the main issues in planned maintenance is that the subjects who participate in the maintenance process provide an insufficient level of information, hence hindering a complete application and verification of maintenance-planning principles and algorithms. In this case, the dataset—which is inserted into the app after every inspection of intervention, to update the state of conservation of the involved components—is optimized, as it only requires information that is easily available to maintenance operators and does not require any qualitative judgment; the evaluation of the degradation index directly derives from the indication of the decay typology and extension.

**Table 6.** Overview of the parameters in the Master Database with their visibility and editability conditions in the Branch to Core App, and the related conditions for their input.

| Level | Master Database Parameter | Visibility | Editability | Input Conditions/Options |
|---|---|---|---|---|
| 1 | Technological Unit Class | No | No | - |
| 2 | Technological Unit | No | No | - |
| 3 | Technical Element Class | Yes | No | - |
| 4 | Technical Element ID | Yes | No | - |
| 5 | Material | Yes | No | - |
| | Exposure | Yes | No | - |
| | Position | Yes | No | - |
| | Surface Area | Yes | No | - |
| 6 | Decay Typology $n$ | Yes | Yes | All decay typologies |
| | Decay Extension $n$ | Yes | Yes | Positive number < Surface Area |
| | Decay Severity | No | Automatic update | - |
| | Decay Burden | No | Automatic update | - |
| | Degradation Index | No | Automatic update | - |
| 7 | Inspection typology | Yes | Yes | Maintenance Inspection Cards |
| | Inspection description | Yes | Automatic update | - |
| | Inspection frequency | Yes | Automatic update | - |
| | Intervention typology | Yes | Yes | Maintenance Intervention Cards |
| | Intervention description | Yes | Automatic update | - |
| | Intervention frequency | Yes | Automatic update | - |
| | Last inspection date | Yes | Yes | DD/MM/YYYY format |
| | Last intervention date | Yes | Yes | DD/MM/YYYY format |
| | Inspection cost | No | Automatic update | - |
| | Maintenance cost | No | Automatic update | - |

## 4. Implementation of the Methodology

As mentioned in the introduction, the methodology was implemented and tested in the framework of the WARMEST project, specifically on the cloister of the Church of Santa Croce, Florence, in Italy. As shown in Figure 2, the Branch to Core App was also experimented with in another site of the project, Patio de Los Leones in the Alhambra complex, in Granada, Spain; however, in that case, the method was not fully developed, so the results of that experimentation are not included in this paper.

### 4.1. Construction of the Master Database

The structuring of the Master Database followed the method described in Section 3.1. With reference to the breakdown entries suggested by the UNI 8290 norm for Levels 1, 2, and 3, which are described and fully listed in Section 2.1, the articulation of the main Levels was modified, in accordance with the typology of the artifact. Indeed, since the analyzed architectural artifact is a cloister, systems are not considered, and a major focus was placed on the differentiation and variety of pavements, roof coverings, and doors and windows.

For the first three levels, the following entries were used:

- Level 1—Structures, Claddings, Doors and windows, Roof coverings, Valuable elements, Pavements;
- Level 2—Aboveground masonry structures, Roof structures, External claddings, Doors and windows, Sloped coverings, Stone elements, Primary elements, External pavements, Internal pavements;
- Level 3—Stone walls, Stone vaults, Wood structures, Plaster, Doors with wings, Gutters and drainpipes, Composite panels for roof ventilation, Waterproofing layer with tiles, Exposed grey sandstone columns, Cornices, Exposed grey sandstone portals, Exposed grey sandstone arches, Exposed grey sandstone emblem, Exposed grey sandstone masonry, Terracotta cladding, Stone cladding.

Concerning the additional levels, which are entirely defined within this methodology, the following options were established in order to characterize all the components of the cloister:

- Technical element: Perimetral wall, Trabeation wall, Rib vault, Primary beam, Truss beam, Perimetral wall plaster, Vault plaster, Entrance gate, One-wing door, Two-wing door, Glazed door, Watershed gutter, Composite panels, Tiles, Column, Cornice, Gate, Arch, Emblem, Stone, Small stone wall, Terracotta pavement, Stone pavement;
- Material: Sandstone, Wood, Cement lime, Iron, Wood and glass, Steel, Terracotta;
- Exposition: North-East, North-West, South-East, South-West;
- Position: Ground floor, First floor, Trabeation floor, Roof.

Levels 4 and 5, illustrated above, with the addition of surface area, which took variable numerical values, stand as the "fixed" part of the decomposition and the restriction of their domains is the consequence of the analysis of the specific characteristics of the cloister and its components. Level 6, associated with the evaluation and quantification of the decay condition of each of the components—which is updated over time—is not restricted to a sub-domain of the possibilities listed in the previous paragraphs for stone, wood, and metal components. Of course, each of the components may suffer from scarcely predictable and variable forms of decay over time, in addition to the ones observed at the time (2017–2018) in which this analysis was carried out. Therefore, the sets of decay typologies coincide with the ones from Section 3.1. Figure 4 reports an excerpt from the Master Database realized for the Cloister of Santa Croce.

| | A | B | C | D | E | F | G | H | I |
|---|---|---|---|---|---|---|---|---|---|
| 1 | ID | Technological unit class | Technological unit | Technical element class | Technical element | Material | Exposure | Location | Surface Area [m²] |
| 2 | 01.01.01.01 | Structures | Aboveground masonry structure | Stone walls | Perimetral wall | Sandstone | SW | Ground floor | 231.34 |
| 3 | 01.01.01.02 | Structures | Aboveground masonry structure | Stone walls | Perimetral wall | Sandstone | SE | Ground floor | 284.29 |
| 4 | 01.01.01.03 | Structures | Aboveground masonry structure | Stone walls | Perimetral wall | Sandstone | NE | Ground floor | 234.76 |
| 5 | 01.01.01.04 | Structures | Aboveground masonry structure | Stone walls | Perimetral wall | Sandstone | NO | Ground floor | 283.27 |
| 6 | 01.01.01.05 | Structures | Aboveground masonry structure | Stone walls | Perimetral wall | Sandstone | SW | First floor | 248.53 |
| 7 | 01.01.01.06 | Structures | Aboveground masonry structure | Stone walls | Perimetral wall | Sandstone | SE | First floor | 308.14 |
| 8 | 01.01.01.07 | Structures | Aboveground masonry structure | Stone walls | Perimetral wall | Sandstone | NE | First floor | 256.27 |
| 9 | 01.01.01.08 | Structures | Aboveground masonry structure | Stone walls | Perimetral wall | Sandstone | NW | First floor | 308.14 |
| 10 | 01.01.01.09 | Structures | Aboveground masonry structure | Stone walls | Trabeation wall | Sandstone | SW | Trabeation floor | 55.54 |
| 11 | 01.01.01.10 | Structures | Aboveground masonry structure | Stone walls | Trabeation wall | Sandstone | SE | Trabeation floor | 71.44 |
| 12 | 01.01.01.11 | Structures | Aboveground masonry structure | Stone walls | Trabeation wall | Sandstone | NE | Trabeation floor | 55.54 |
| 13 | 01.01.01.12 | Structures | Aboveground masonry structure | Stone walls | Trabeation wall | Sandstone | NW | Trabeation floor | 71.42 |
| 14 | 01.01.03.01 | Structures | Aboveground masonry structure | Stone vaults | Rib vault | Sandstone | - | Trabeation floor | 16.48 |
| 15 | 01.01.03.02 | Structures | Aboveground masonry structure | Stone vaults | Rib vault | Sandstone | - | Trabeation floor | 16.48 |
| 16 | 01.01.03.03 | Structures | Aboveground masonry structure | Stone vaults | Rib vault | Sandstone | - | Trabeation floor | 16.48 |
| 17 | 01.01.03.04 | Structures | Aboveground masonry structure | Stone vaults | Rib vault | Sandstone | - | Trabeation floor | 16.48 |
| 18 | 01.01.03.05 | Structures | Aboveground masonry structure | Stone vaults | Rib vault | Sandstone | - | Trabeation floor | 16.48 |
| 19 | 01.01.03.06 | Structures | Aboveground masonry structure | Stone vaults | Rib vault | Sandstone | - | Trabeation floor | 16.48 |
| 20 | 01.01.03.07 | Structures | Aboveground masonry structure | Stone vaults | Rib vault | Sandstone | - | Trabeation floor | 16.48 |
| 21 | 01.01.03.08 | Structures | Aboveground masonry structure | Stone vaults | Rib vault | Sandstone | - | Trabeation floor | 16.48 |
| 22 | 01.01.03.09 | Structures | Aboveground masonry structure | Stone vaults | Rib vault | Sandstone | - | Trabeation floor | 16.48 |
| 23 | 01.01.03.10 | Structures | Aboveground masonry structure | Stone vaults | Rib vault | Sandstone | - | Trabeation floor | 16.48 |
| 24 | 01.01.03.11 | Structures | Aboveground masonry structure | Stone vaults | Rib vault | Sandstone | - | Trabeation floor | 16.48 |
| 25 | 01.01.03.12 | Structures | Aboveground masonry structure | Stone vaults | Rib vault | Sandstone | - | Trabeation floor | 16.48 |
| 26 | 01.01.03.13 | Structures | Aboveground masonry structure | Stone vaults | Rib vault | Sandstone | - | Trabeation floor | 16.48 |

**Figure 4.** Excerpt from the Master Database of the Cloister of Santa Croce.

*4.2. Decay Evaluation Algorithm*

For each of the components, the degradation index was calculated through the data collected on the first survey of the architectural artifact, in terms of decay typologies and related extensions. Based on the coefficients defined in the methodology and illustrated in Sections 3.2.1 and 3.2.2, and relating the extensions of the single typologies of decay to the total area of the component in every case, it was possible to determine the degradation index.

The following Table 7 reports a focus on the evaluation of the degradation index for the columns; the layout is slightly different from that of the Master Database, showing fixed decay typologies on the columns and extensions below each of them. This is mainly due to the restriction of the exemplified set to columns—which share the same typologies of decay—in order to allow easier visualization of all the related parameters; however, all the entries are the same as in the Master Database.

**Table 7.** Synthetic table with the elaboration of surface area, decay extension, and degradation index values for Columns 1–18 of the Cloister of Santa Croce.

| ID | Surface Area [m²] | Chromatic Alteration [m²] | Exfoliation [m²] | Powdering [m²] | Erosion [m²] | Detachment [m²] | Degradation Index |
|---|---|---|---|---|---|---|---|
| 01.11.01.01 | 2.54 | 0.64 | 0.64 | - | 0.05 | - | 0.23 |
| 01.11.01.02 | 2.54 | - | 0.43 | - | 0.13 | 0.84 | 0.32 |
| 01.11.01.03 | 2.54 | - | 0.43 | - | 0.13 | 0.84 | 0.27 |
| 01.11.01.04 | 2.54 | 0.64 | 0.84 | - | 0.03 | - | 0.29 |
| 01.11.01.05 | 2.54 | 0.64 | 0.43 | - | 0.03 | - | 0.15 |
| 01.11.01.06 | 2.54 | - | - | - | 0.38 | 0.43 | 0.21 |
| 01.11.01.07 | 2.54 | - | - | - | 0.38 | 0.43 | 0.23 |
| 01.11.01.08 | 2.54 | - | - | - | 0.13 | - | 0.05 |
| 01.11.01.09 | 2.54 | - | - | - | 0.13 | - | 0.05 |
| 01.11.01.10 | 2.54 | - | 0.64 | 0.25 | - | - | 0.27 |
| 01.11.01.11 | 2.54 | - | 0.64 | 0.25 | 0.64 | 0.20 | 0.23 |
| 01.11.01.12 | 2.54 | - | 0.64 | - | 0.13 | - | 0.27 |
| 01.11.01.13 | 2.54 | - | 1.27 | - | 0.76 | 0.41 | 0.25 |
| 01.11.01.14 | 2.54 | - | 0.64 | - | 0.25 | 0.13 | 0.21 |
| 01.11.01.15 | 2.54 | - | 0.64 | - | 0.13 | 0.13 | 0.20 |
| 01.11.01.16 | 2.54 | - | 0.64 | - | - | 0.38 | 0.34 |
| 01.11.01.17 | 2.54 | - | - | - | 0.13 | - | 0.05 |
| 01.11.01.18 | 2.54 | - | 0.43 | - | 0.25 | - | 0.21 |

As illustrated in Sections 3.2.1 and 3.2.2, the decay typologies in Table 7 are associated with the following values of decay severity ($K_D$) and decay burden ($K_B$):

- Chromatic Alteration—$K_D = 1$, $K_B = 0.18$;
- Exfoliation—$K_D = 4$, $K_B = 1.71$;
- Powdering—$K_D = 3$, $K_B = 0.76$;
- Erosion—$K_D = 3$, $K_B = 0.76$;
- Detachment—$K_D = 3$, $K_B = 0.76$.

*4.3. Use of the Branch to Core App*

The data collection process took place in a limited time frame (2017–2018), so the evolution of the state of decay was not tracked; rather, a single-time condition was recorded in this implementation. Of course, the purpose of the Branch to Core App is extended beyond that: its full potential and benefits are realized in the framework of the update of the condition of components over time, through an easy modality of overwriting data. However, the Branch to Core App was used for the initial inspection of the decay conditions of the components of the Cloister of Santa Croce in order to test its compatibility in the process and its full capability of inputting data and filling in the Master Database directly from a mobile device. The layout of the app for the Cloister of Santa Croce, developed with the same graphics and settings as the prototype shown in Section 3.3, can be seen in Figure 3.

Hence, its integration in the process was actuated as follows: when performing the initial inspection of the building, the data regarding the organization and the subdivision of the building into its single components according to the WBS structure, in addition to the fixed data regarding their material, position, and surface area, were directly input on the Master Database through Microsoft Excel; instead, the data regarding the decay condition

(decay typologies, decay areas), shown in Section 4.2, were input on the Master Database from a mobile device using the Branch to Core App. Figure 5 reports some excerpts of the WBS realized for the Cloister of Santa Croce and data input from the app.

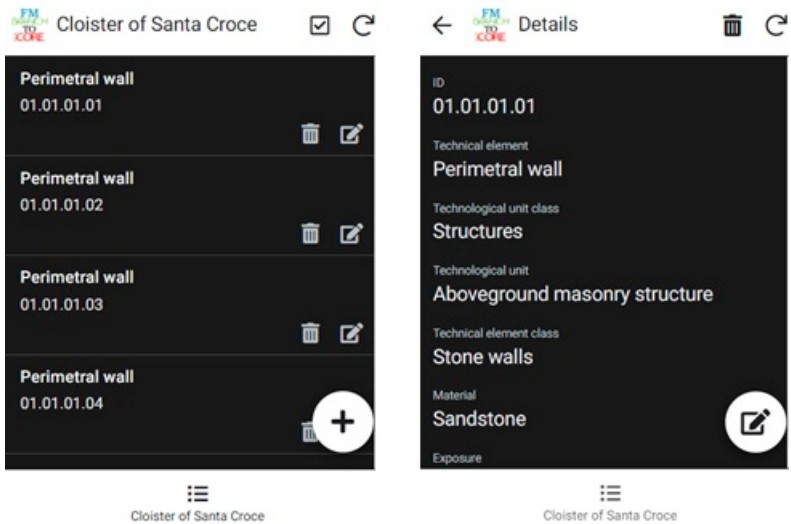

**Figure 5.** Screenshots from the utilization of the app for the Cloister of Santa Croce.

## 5. Discussion and Conclusions

Since app testing was related to the data input of decay parameters, it did not involve the quantification of the economic benefits deriving from its implementation in terms of the reduction of maintenance costs. However, the feedback from the administration of the Basilica of Santa Croce proved that the developed tool fulfilled the technical needs that were initially declared. During the testing phase, data input proved to be a smooth and clear process for the maintenance operators, and the database could be easily interpreted by the technical bureau. Hence, the app matched the requirements at the basis of its conception.

The information paradigm of our era is constantly requiring the implementation of systems for the centralized collection of information, to be elaborated through increasingly accurate methodologies and decision support systems for identifying and implementing the most suitable available choices. At the same time, these advances are calling for more and more pervasive integration of smart technologies, sensors, and devices to ensure continuous and reliable sharing of data between all the involved subjects. The peculiar combination of the fulfillment of these two needs—data processing and data collection framework—in the proposed methodology derives from an approach of *soft* additions to the paradigm of building maintenance, due to the easily recognizable delay in a diffuse assumption and incorporation of technologically complex solutions in the building sector. Meanwhile, in opposition to this tendency, BIM-based approaches [43], IoT devices [44], and digital twins [45] are now gaining more ground in relation to maintenance; however, there are still significant barriers to the adoption of these opportunities in smaller projects and for smaller firms, due to the need for trained personnel and higher technological costs. Moreover, in many contexts, there is still resistance toward radical changes from the traditional approach, even for valuable architectures.

In this sense, this methodology follows an opposite direction, compared to the main-stream tendency, being based on simple tools and on easily implementable computations; however, for this reason, it stands even more as an innovation, as it fills the gap between the practice and use of complex technological systems, and the consistent share of the built environment that is not prone to avail of them. Its users can immediately apply it to their framework of building condition monitoring, smoothing traditional procedures by introducing a reliable and simple method for the assessment of building conditions, as well as a combination of well-known and diffuse software tools to prevent data loss and generate small-scale interoperability concerning inspection results.

However, these components and tendencies are not structurally incompatible with each other: the integration of more complex and articulate IT tools—such as BIM interfaces and models—with this methodology can improve the visualization and consultation features of the output data. An example of convergent development could be the realization of smart maintenance-oriented BIM models, that would facilitate the visualization of degradation aspects—decay severity, typology of decay, and degradation index—for an entire building, by attributing univocal colorimetric and graphic parameters. Another upgrade might be related to the accuracy of decision support systems concerning maintenance plans: deep learning approaches can provide valid support to the semi-automatization of maintenance planning, but their development and training require the availability of structured datasets. Thus, the Branch to Core methodology stands as a valid foundation for this kind of addition.

These perspectives of further developments allow for the envisaging of a multi-level future asset of this methodology: the simplicity of its current state enables any typologies of technicians and professionals to include and apply it to their management framework, while the incorporated presence of more complex features—whose use requires more structured know-how and capacities—will also provide greater benefits to the higher-skill-level work teams to which they are destined.

**Author Contributions:** Conceptualization, G.A.; methodology, G.A. and A.P.; software, A.P.; writing—original draft preparation, A.P.; writing—review and editing, G.A. and A.P.; funding acquisition, G.A.; visualization, A.P.; supervision, G.A.; project administration, G.A. All authors have read and agreed to the published version of the manuscript.

**Funding:** This research was funded by the European Union Horizon 2020 Research and Innovation Program "WARMEST"—or "loW Altitude Remote sensing for the Monitoring of the state of cultural hEritage Sites: building an inTegrated model for maintenance"—through the Marie Skłodowska-Curie Grant Agreement no. 777981.

**Institutional Review Board Statement:** Not applicable.

**Informed Consent Statement:** Not applicable.

**Data Availability Statement:** The data presented in this study are available on request from the corresponding author. The data are not publicly available due to the decision of keeping technical data confidential in the WARMEST project.

**Acknowledgments:** The authors thank Arch. Claudia Mariaserena Parisi for her contribution to the early development of the methodology.

**Conflicts of Interest:** The authors declare no conflict of interest.

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
