# Peer review of "A Simplified Facility Management Tool for Condition Assessment through Economic Evaluation and Data Centralization: Branch to Core"

_sustainability, doi:10.3390/su15086418_

Round 1

Reviewer 1 Report

This paper presents a framework for facility management, involving data collection, systematization, and elaboration on a mobile app for automatic data collection. The proposed methodology has been tested on a case study. The following are some concerns for this reviewer.

-          The title of the paper is about facility management; however, facility management is very broad subject that entails many developments. The focus of the paper was only on a condition assessment as per the understanding of this reviewer. Even though, it was not introduced clearly and thoroughly.

-          Also, the title is talking about a novel methodology, however, it was not clear for this review what was novel in the presented development.

-          The paper lacks presenting a thorough review of the literature, especially the related work.

-          The paper is presenting thoughts rather than deep and rigorous development.

-          More details are needed specially for assessing the condition part.

-          The paper should clearly state the objective(s) of the presented development.

-          Assessing the condition of a given asset entails creating the WBS and then assessing the elements at the lower level and then rolling up for the upper components. How this was done in this paper.

-          The conclusion section is more general speaking rather than conclusions based on the presented study.

Author Response

Reviewer: "The title of the paper is about facility management; however, facility management is very broad subject that entails many developments. The focus of the paper was only on a condition assessment as per the understanding of this reviewer. Even though, it was not introduced clearly and thoroughly"

Response: The focus on Condition Assessment has now been more clearly specified, both in the title and in a dedicated paragraph.

Reviewer: "Also, the title is talking about a novel methodology, however, it was not clear for this review what was novel in the presented development."

Response: the word "novel" has been removed, as it indeed gave a wrong perception of the fine-tuned tool.

Reviewer: "The paper lacks presenting a thorough review of the literature, especially the related work."

Response: This comment has been interpreted as a request for a specific literature review on condition assessment tools. A paragraph was added to the "Materials and methods".

Reviewer: "The paper is presenting thoughts rather than deep and rigorous development."

Response: The research work originated from a specific request from the administration of Basilica di Santa Croce, and hence the presented deficiencies in the field of management derived from their specific needs. 

Reviewer: "

More details are needed specially for assessing the condition part."

Response: The adopted parameters synthesize all decay-related aspects: this setting derives from a recognized formulation in the literature (Degradation Index), while its specification simply allowed estimating the parameters without excessive technical knowledge by the operators.

Reviewer: "

The paper should clearly state the objective(s) of the presented development."

Response: A specific part was added to declare the derivation of the tool's purpose from the administration of Basilica di Santa Croce.

Reviewer: "

Assessing the condition of a given asset entails creating the WBS and then assessing the elements at the lower level and then rolling up for the upper components. How this was done in this paper."

Response: The purpose was not to define the state of conservation of the asset as a whole but of the single components. The usefulness of the WBS is to group the components into homogeneous categories, to better organize the data.

Reviewer: "The conclusion section is more general speaking rather than conclusions based on the presented study."

Response: The conclusions were expanded with a focus on the results of the research work.

Reviewer 2 Report

- It's not clear what the authors' contributions are. Is this a literature review and creation of an app?

The authors present an interesting study on FM and propose an easy application for tracking it.

- What are the benefits from using this app? The benefits need to be shown in a quantitative manner.

- The case study is not clear. Is the objective of this app filling in a simple form? How is this innovative?

- Line 443: why only "4 + 2n"? what does this symbolize?

- The authors mention several parameters in the literature part. How are these reflected in the app?

- The discussion needs more work to showcase the importance of the app.

Author Response

Reviewer: "It's not clear what the authors' contributions are. Is this a literature review and creation of an app?"

Response: A diagram was added before the outline of the research items, to further clarify the performed work and the direct entailment of the parameter definition and WBS structure expansion for condition assessment and simplified data centralization.

Reviewer: "What are the benefits from using this app? The benefits need to be shown in a quantitative manner."

Response: As detailed in the diagram, the app does not include by itself making decisions on the maintenance planning, therefore quantitative benefits are not directly evaluated in this phase. The administration and staff of Basilica di Santa Croce expressed positive feedback on the easier implementation of the process, but this could naturally only be communicated qualitatively.

Reviewer: "The case study is not clear. Is the objective of this app filling in a simple form? How is this innovative?"

Response: The app is by itself the final phase of the process; in other words, it is simple to fill in the form through the app because all the parameters have been defined in relation to directly observable phenomena and characteristics of the building components. However, as stated before, the workflow has been synthesized in a diagram.

Reviewer: "Line 443: why only "4 + 2n"? what does this symbolize?"

Response: Sorry for the omission, it was not explained clearly. Now this has been fixed.

Reviewer: "The authors mention several parameters in the literature part. How are these reflected in the app?"

Response: Out of Decay Indices, the Degradation Index was selected and further specified: it is indeed crucial for decay evaluation in the original methodology.

Reviewer: "The discussion needs more work to showcase the importance of the app."

Response: As of now, the only available data consist of feedback from its users, that is the ones from the administration of Basilica di Santa Croce. Further applications and wider diffusion of the methodology will allow analyzing its benefits.

Author Response

Thank you for the generally positive comment on our article, we strongly appreciate it and have now tried to solve all the issues that you have noticed.

Reviewer: "The keywords should also include international/national standards, and decay evaluation."

Response: Several standards were included in the Materials and methods; out of them, the most relevant one was definitely ISO 12006-2:2015, which was then added to the keywords together with "decay evaluation", as requested.

Reviewer: "As shown in Table 7, maintenance operators only have to fill 4 + 2n entries,…….Where is table 7? I could not locate it in the manuscript."

Response: There were some misalignments in the numbers and distribution of figures and tables, which have now been fixed.

Reviewer: "Each figure and table have a descriptive title that describes its contents in detail. However, the numbering need to be checked and updated appropriately. And some are not explained in the text."

Response: As above, now they have all been fixed. Thank you for the notice.

Reviewer: "There are figures that should be in table form (e.g., figures 1 and 4)."

Response: That is correct, this choice was mainly due to aesthetical reasons, but now all figures deriving from tables have been transformed into tables.

Reviewer: "Figure 1 should be changed to proper WBS (i.e., hierarchy). It looks more like a table to me."

Response: The employed structure follows the model of the Italian standard UNI 8290, as reported just before the table.

Round 2

Reviewer 1 Report

This revised version of the manuscript has greatly improved, the paper still lacking some focus. For example, in section 2.3, the authors talk about building performance evaluation methodologies in general including certification, energy performance, etc. However, the scope of this research is buildings condition assessment not buildings ratings or performance evaluation. 

Author Response

We acknowledge that it could have been misleading; hence, the title of the paragraph was changed to "Methods for the evaluation of building condition"; moreover, concerning building certification protocols, a sentence was added to clarify the reason behind their inclusion in the dissertation.

"The purpose of building certification protocols is inherently different from building condition assessment methods, as they are intended to assess the general state of the building for reasons related to energy performance; however, the parameters and scales adopted to evaluate this specific aspect within them can be taken as a reference among qualitative approaches for the determination of building condition."

Reviewer 2 Report

The authors have taken care of all comments. Minor changes cna be made to the grammar to improve readability in some instances.

One minor suggestion below.

Figure 1 and the newly added parts surrounding it: it's not clear whether this refers to previous work or this research. for example, line 286 "the research work..". Are you referring to previous research or this paper? Please clarify that so the readers can understand if you're referring to the gap in literature or to your own work.

Author Response

Thank you for the suggestion, as we had not noticed this possibility of generating confusion in that paragraph. It was referred to the research work related to this paper. I have rephrased this, to make it as clear as possible now, I hope this solves it:

"Figure 1 graphically outlines the research work carried out for the development of the Branch to Core method and app, presented for the first time in this paper. As shown here, ..."